# Degradation-Sensitive Health Indicator Construction for Precise Insulation Degradation Monitoring of Electromagnetic Coils

**DOI:** 10.3390/e25091354

**Published:** 2023-09-19

**Authors:** Yue Sun, Kai Wang, Aidong Xu, Beiye Guan, Ruiqi Li, Bo Zhang, Xiufang Zhou

**Affiliations:** 1Key Laboratory of Networked Control Systems, Chinese Academy of Sciences, Shenyang 110169, China; sunyue@sia.cn (Y.S.); xad@sia.cn (A.X.); guanbeiye@sia.cn (B.G.); liruiqi1@sia.cn (R.L.); zhangbo1@sia.cn (B.Z.); zhouxiufang@sia.cn (X.Z.); 2Shenyang Institute of Automation, Chinese Academy of Sciences, Shenyang 110169, China; 3Institutes for Robotics and Intelligent Manufacturing, Chinese Academy of Sciences, Shenyang 110169, China; 4University of Chinese Academy of Sciences, Beijing 100049, China

**Keywords:** electromagnetic coils, insulation degradation monitoring, high-frequency electrical response parameters, health indicator, threshold setting

## Abstract

Electromagnetic coils are indispensable components for energy conversion and transformation in various systems across industries. However, electromagnetic coil insulation failure occurs frequently, which can lead to serious consequences. To facilitate predictive maintenance for industrial systems, it is essential to monitor insulation degradation prior to the formation of turn-to-turn shorts. This paper experimentally investigates coil insulation degradation from both macro and micro perspectives. At the macro level, an evaluation index based on a weighted linear combination of trend, monotonicity and robustness is proposed to construct a degradation-sensitive health indicator (DSHI) based on high-frequency electrical response parameters for precise insulation degradation monitoring. While at the micro level, a coil finite element analysis and twisted pair accelerated degradation test are conducted to obtain the actual turn-to-turn insulation status. The correlation analysis between macroscopic and microscopic effects of insulation degradation is used to verify the proposed DSHI-based method. Further, it helps to determine the threshold of DSHI. This breakthrough opens new possibilities for predictive maintenance for industrial equipment that incorporates coils.

## 1. Introduction

As an important energy conversion component for most components, such as equipment and large equipment, the performance of electromagnetic coils will inevitably deteriorate during industrial production due to long-term operation in harsh environments, which may cause serious economic losses [1]. Therefore, on-line monitoring of electromagnetic coils plays an important role in avoiding accidental shutdown of the equipment and reducing economic losses [2,3]. On-line monitoring methods can be further divided into turn-to-turn short fault detection and turn-to-turn insulation degradation monitoring methods. Currently, most on-line monitoring methods fall into turn-to-turn short fault detection. The classical methods include motor current signature analysis (MCSA) [4,5], negative-sequence-current-component-based methods [6], Park’s vector-based methods [7], and motor magnetic field signals-based methods [8,9], etc. Deep learning methods have also been used for turn-to-turn short circuit fault detection, including convolutional neural network-based methods [10,11], deep auto-encoder-based methods [12], etc. The application premise of the turn-to-turn short circuit fault detection methods is that a turn-to-turn short circuit fault has occurred. However, a turn-to-turn short circuit fault can rapidly spread throughout a coil and result in its complete failure. In the case of a 15 KW induction motor, it takes less than 2 s from the start of a turn-to-turn short circuit fault to the complete failure of the winding [13]. Therefore, compared with turn-to-turn short fault detection methods, turn-to-turn insulation degradation methods, which can monitor insulation status prior to the formation of turn-to-turn shorts, have more contributions to the realization of predictive maintenance of equipment and the formulation of overall maintenance plans.

The existing typical turn-to-turn insulation degradation monitoring methods are summarized as follows. Werynski et al. [14] and Perisse et al. [15] have performed accelerated aging tests on twisted pair and demonstrated that the turn-to-turn capacitance increases as the breakdown voltage decreases. Savin et al. [16] also found a correlation between turn-to-turn capacitance and partial discharge initial voltage (PDIV). Thus, these studies have proved that turn-to-turn capacitance is an effective indicator of insulation degradation. Unfortunately, it is not feasible to measure turn-to-turn capacitance on-site. Therefore, the influence of turn-to-turn capacitance on the whole winding behavior is studied for the purpose of implementing online insulation degradation monitoring. Perisse et al. [15] developed a complex equivalent circuit model to quantify the relationship between winding impedance and turn-to-turn capacitance. Further, they proposed an online insulation degradation monitoring method by measuring the phase shift and the resulting magnetic field by injecting low-level, high-frequency signals at the coil resonant frequency. Younsi et al. [17] used a High-Sensitivity-Current Transformer (HSCT) to monitor the capacitance, AC insulation resistance and dissipation factor on the stator windings of a wire-wound motor. The decrease in capacitance with aging of the insulation was demonstrated by simulated high-temperature ageing tests. However, no winding-level life tests were performed to verify the proposed on-line insulation degradation monitoring methods. Neti et al. [18] studied winding high-frequency electrical response (impedance) by building a high-frequency parametric winding model and simulating the motor winding impedance response of the model by modification of turn-to-turn capacitance values manually. In detail, winding insulation aging is simulated by connecting variable capacitors and resistors in parallel with sections of motor coils. They demonstrated that the motor coil impedance response changes with resistors and capacitors. However, no winding life tests were performed, so the real correlation between motor coil impedance response and turn-to-turn capacitance is still not clear. Jordan et al. [19], at the University of Maryland, conducted an accelerated degradation test on a hand-wound coil and measured its impedance spectrum. Using the Spearman correlation coefficient [20], impedance measurements at specific frequencies were utilized for insulation degradation monitoring. However, correlation between coil impedance response and the actual turn-to-turn insulation health status has not been studied. Consequently, challenges persist in insulation degradation monitoring, such as how to determine the on-line health monitoring parameter failure threshold. In addition, only the monotonicity index is considered to construct the on-line monitoring parameter based on impedance measurements in [20], while other significant indexes, such as trend and robustness, are neglected.

To address this issue, this paper takes a small commercial transformer as the research object and carries out insulation degradation monitoring research simultaneously at the macro and micro levels. On the one hand, an evaluation index based on a weighted linear combination of trend, monotonicity and robustness is proposed to construct a DSHI based on impedance evolution characteristics for precise insulation degradation monitoring. On the other hand, the transformer temperature field distribution is obtained by FEM under the accelerated test condition, and thus an accelerated aging test on twisted pairs were designed and performed to explore the correlation between the health indicator of the proposed method and actual turn-to-turn insulation health status, which helps to provide engineering guidance for determining the health indicator failure threshold.

The organizational structure of this paper is as follows. Section 2 describes the precise insulation degradation monitoring method of electromagnetic coils proposed in this paper. Section 3 describes the experimental setup and result analysis. In Section 4, the conclusion of this paper is provided.

## 2. Insulation Degradation Monitoring Method of Electromagnetic Coils

The methodology framework of coil precise insulation degradation monitoring is shown in Figure 1. The high-frequency electrical response parameters are obtained to track the insulation degradation of electromagnetic coils. The preliminary physical health indicators and virtual health indicators are constructed based on the high-frequency electrical parameters. Then, the degradation-sensitive health indicator (DSHI) is determined based on the new health indicator evaluation index and criteria. Finally, the coil insulation degradation state is accurately evaluated based on the DSHI and actual turn-to-turn insulation health status. The specific process is as follows:

### 2.1. Collection of High-Frequency Electrical Response Parameters

High-frequency electrical parameter analysis is an effective method for monitoring the degradation of electromagnetic coils. Impedance, resonant frequency, and parasitic capacitance are common high-frequency electrical response parameters. In general, high-frequency electrical response parameters can be generated through accelerated tests or historical data. During the accelerated test, high-frequency electrical responses collection equipment is used to inject small-amplitude high-frequency signals into the coil to be tested and collect high-frequency electrical response impedance signals. Data set Z is constructed under multiple frequencies of the coil
$$Z=\left(\begin{array}{cccccc} Z(t_{0},f_{1}) & Z(t_{1},f_{1}) & \cdots & Z(t_{k},f_{1}) & \cdots & Z(t_{T},f_{1})\\ Z(t_{0},f_{2}) & Z(t_{1},f_{2}) & \cdots & Z(t_{k},f_{2}) & \cdots & Z(t_{T},f_{2})\\ \vdots & \vdots & \cdots & \vdots & \cdots & \vdots \\ Z(t_{0},f_{N}) & Z(t_{1},f_{N}) & \cdots & Z(t_{k},f_{N}) & \cdots & Z(t_{T},f_{N}) \end{array}\right)$$
where the coil aging cycle is denoted as $t=[t_{0},t_{1},\cdot \cdot \cdot ,t_{k},\cdot \cdot \cdot ,t_{T}]$; $t_{0}$ denotes the coil health cycle; $t_{T}$ denotes that T cycles were experienced; $f=[f_{1},f_{2},\ldots ,f_{N}]$ denotes the collection frequency of the coil electrical response parameters; N is the number of collection frequencies injected; and $Z(t_{T},f_{N})$ denotes the electrical parameters at the T-th cycle, frequency. $Z(t_{k})=[Z(t_{k},f_{1}),Z(t_{k},f_{2}),\ldots ,Z(t_{k},f_{N})]$ denotes the full frequency range impedance spectrum for the k-th cycle.

### 2.2. Construction of Preliminary Health Indicators

In this paper, root mean square (RMS) and kurtosis (K), which are effective time-domain features in fault diagnosis and health management, are considered. However, in the process of electromagnetic coil insulation degradation, the degradation features are usually nonstationary and nonlinear. For this reason, the time-frequency domain features of fuzzy entropy (FE) and wavelet packet node energy (WPNE) are also extracted. Such physical quantities that characterize equipment degradation are generally referred to as physical health indicators. At the same time, information fusion algorithms, such as Principal Component Analysis (PCA) and Mahalanobis–Taguchi system (MTS) are considered for feature dimension reduction to avoid information redundancy and model training overfitting. PCA [21] and MTS [22,23] are both indicators of equipment degradation after information fusion of physical health indicators. In this paper, they are referred to as virtual health indicators.

The above physical health indicators and virtual health indicators are used as preliminary health indicators. The calculation equations of the preliminary health indicators are shown in Table 1. In the table, both RMS ($HI_{rms}$) and K ($HI_{k}$) are calculated based on $Z(t_{k})$, where $\bar{Z(t_{k})}$ is the mean value of the impedance spectrum for the k-th cycle $Z(t_{k})$; and FE is used to measure the similarity of two sequences. In the calculation process of FE ($HI_{fe}$), the phase space is reconstructed based on the impedance spectral sequence $Z(t_{k})$ and related parameters(m,n,r), while the affiliation function is introduced to calculate the mean values $\phi^{m}(n,r)$, $\phi^{m+1}(n,r)$ of all affiliations except itself in m and m+1 dimensions, respectively. $\phi^{m}(n,r)$ is the similarity function of the two reconstructed m-dimensional vectors, where n is the boundary gradient, and r is the similarity tolerance; the WPNE ($HI_{wpne_{i}}$) is obtained by wavelet decomposition and reconstruction of $Z(t_{k})$ to obtain the reconstruction coefficient $z_{i}$ of the i-th node; P is the eigenvector matrix corresponding to the covariance of the impedance spectral sequence $Z(t_{k})$; MTS uses orthogonal table and signal-to-noise ratio to optimize features and calculate the MD value ($HI_{MD_{j}}$) of the selected features. Among them, $S_{j}$ denotes the standardized matrix of the j-th characteristic parameters of the impedance spectrum $Z(t_{k})$, corr is the correlation coefficient matrix, and p is the number of characteristic parameters.

### 2.3. Determination of DSHI

The determination process of the DSHI based on the above preliminary health indicators $HI_{pre}=[HI_{rms},HI_{k},HI_{fe},HI_{wpne_{i}},HI_{pca},HI_{MD_{j}}]$ is shown in Figure 2. The DSHI evolution trend can characterize the degradation state of electromagnetic coils.

Since the coil insulation degradation layer will gradually become thinner during the insulation degradation process, indicating that the insulation degradation is an irreversible process, the health indicator reflecting coil insulation degradation should have monotonic characteristics. With the increase of the aging cycle, the coil insulation degradation will also change, so the health indicator and the aging time should have a certain correlation, which is also called the trend. In addition, the health indicator should have good anti-interference ability for outliers, that is, it has a certain robustness. In this paper, the evaluation indexes, such as trend, monotonicity and robustness based on trend and residual [24,25,26] are used to select DSHI. For preliminary health indicator series $HI_{pre}$ and time series t, the exponential weighted moving average (EWMA) method is used to decompose the health indicator sequence $HI_{pre}(t_{k})$ at time $t_{k}$ into stationary trend term $HI_{pre}^{T}(t_{k})$ and random residual term $HI_{pre}^{R}(t_{k})$, as shown in (1):(1)$$HI_{pre}(t_{k})=HI_{pre}^{T}(t_{k})+HI_{pre}^{R}(t_{k})$$
where the EWMA calculation process is shown in (2):(2)$$HI_{pre}^{T}(t_{k})=\beta HI_{pre}^{T}(t_{k-1})+(1-\beta )HI_{pre}(t_{k})$$

The equation is generally taken as β≥0.9.

The trend, monotonicity and robustness equations [27] are shown in (3)–(5). Before calculating the three evaluation indexes of the health indicator, the feature sequence needs to be normalized.
(3)$$Tre(HI_{pre},t)=\frac{\left|K\sum_{k}HI_{pre}^{T}(t_{k})t_{k}-\sum_{k}HI_{pre}^{T}(t_{k})\sum_{k}t_{k}\right|}{\sqrt{\left[K\sum_{k}HI_{pre}^{T}{\left(t_{k}\right)}^{2}-{\left(\sum_{k}HI_{pre}^{T}(t_{k})\right)}^{2}\right]\left[K\sum_{k}t_{k}{}^{2}-{\left(\sum_{k}t_{k}\right)}^{2}\right]}}$$
(4)$$Mon(HI_{pre})=\frac{1}{K-1}\left|\sum_{k}\delta \left(HI_{pre}^{T}(t_{k+1})-HI_{pre}^{T}(t_{k})\right)-\sum_{k}\delta \left(HI_{pre}^{T}(t_{k})-HI_{pre}^{T}(t_{k+1})\right)\right|$$
(5)$$Rob(HI_{pre})=\frac{1}{K}\sum_{k}\exp\left(-\left|\frac{HI_{pre}^{R}(t_{k})}{HI_{pre}(t_{k})}\right|\right)$$
where δ is the unit step function, and the specific expression is shown in (6):(6)$$\delta (t)=\{\begin{array}{c} \;1\;,\;t\geq 0\\ \;0\;,\;t<0\; \end{array}$$

The values for each evaluation index of the health indicator are located at [0,1], and the closer to 1, the more sensitive the health indicator is. However, the DSHI cannot be reasonably selected based on a single evaluation index. Therefore, this paper constructs an evaluation index J based on weighted linear combination of trend, monotonicity and robustness, as shown in (7). The closer the value of J is to 1, the more sensitive the corresponding health indicator is. The DSHI is determined according to the above criteria.
(7)$$\begin{array}{l} \underset{HI\in \Omega }{J}=\omega_{1}Tre(HI_{pre},t)+\omega_{2}Mon(HI_{pre})+\omega_{3}Rob(HI_{pre})\\ s.t.\{\begin{array}{c} \omega_{i}>0\\ \sum \omega_{i}=1 \end{array},\;i=1,2,3 \end{array}$$
where $\omega_{1}$, $\omega_{2}$, $\omega_{3}$ are the weights corresponding to each evaluation index. It is known that monotonicity is a prerequisite for the construction of the health indicator, so its weight is the largest. To better track the degradation trend in the health monitoring process, the weight of the trend is set to the second. Since coil insulation degradation monitoring is based on high-frequency electrical response signals in this paper, the signals are not susceptible to noise interference, so the robustness index is provided as the least weight here, i.e., $\omega_{2}>\omega_{1}>\omega_{3}$.

### 2.4. Insulation Degradation Monitoring Based on DSHI

To achieve precise insulation health assessment, this paper constructs a knowledge base to obtain the actual turn-to-turn insulation degradation state. Specifically, the temperature field distribution of the transformer is obtained based on finite element simulation, and then the accelerated aging test of twisted pairs is designed to measure the actual turn-to-turn insulation degradation state index, which is finally used as a benchmark to validate the proposed DSHI-based method. At the same time, the correlation between the DSHI and the actual turn-to-turn insulation state is explored to provide theoretical guidance for determining the DSHI threshold value. The precise insulation degradation monitoring method can provide an early warning of insulation failures and remaining life prediction for coils, which can help reduce the risk of unplanned downtime in industrial systems containing electromagnetic coils.

## 3. Experimental Setup and Result Analysis

In this section, an experimental scheme is proposed, as depicted in Figure 3, which consists of three parts. Initially, the high-frequency response parameters are measured by performing a winding accelerated aging test, and the insulation degradation state of the transformer is monitored based on the existing methods and the DSHI-based method presented in this paper. Subsequently, to verify the accuracy of the method, the breakdown voltage, which reflects the actual insulation degradation, is measured using the temperature mapping method based on finite element simulation and the twisted pair accelerated aging test. This measurement provides a benchmark for the above-mentioned insulation degradation monitoring method. Finally, the DSHI-based method and existing methods are compared with the benchmark to validate the effectiveness of this paper. Among them, the failure threshold setting is the focus of this part.

### 3.1. Winding Accelerated Test and Its High-Frequency Response Measurement and Analysis

#### 3.1.1. Winding Accelerated Test and Its High-Frequency Response Measurement

The winding accelerated aging test platform and high-frequency electric response acquisition system for transformers in this section are shown in Figure 4. A dry-type transformer electromagnetic coil (the insulation material is polyester, and the rated power is 25 W) is selected for accelerated aging experiments under thermal stress. The transformer coil is placed in an 80-degree thermal aging environmental chamber. The input voltage of the transformer is 220 V. The transformer is overloaded to 3.6 times (90 W) the rated power by increasing the load at the output end of the transformer. Then, the transformer coil is removed periodically from the chamber every 8 h for high-frequency electrical response measurement. The accelerated fatigue test procedure is shown in Figure 5. Seven cycles, namely 56 h of the thermally accelerated test, were completed in total. As the coil is removed from the chamber at a high temperature, it must be cooled to room temperature in a constant temperature chamber for 12 h before the high-frequency electrical response parameters can be measured. The termination condition of the accelerated test is that the direct current resistance (DCR) of the transformer coil decreases significantly. According to the measured DCR value, the DCR in the seventh cycle drops significantly to 0.7218 Ω, indicating that the coil has been short-circuited at this time.

After cooling, the high-frequency electrical response signals of the transformer primary coil in the frequency range of 20 Hz–1 MHz will be collected by an E4980A impedance analyzer. Figure 6 shows the resistance and reactance data of the whole life cycle. According to the analysis of the failure mechanism of electromagnetic coils, the reactance only includes the inductive reactance and the capacitive reactance, which is directly affected by the parasitic capacitance. Theoretically, the reactance is the component that directly reflects the degradation of electromagnetic coils. In this paper, the reactance data will be used to analyze the insulation degradation monitoring.

#### 3.1.2. Experiment Results Analysis of the DSHI-Based Method

Based on the reactance data of the whole life cycle of electromagnetic coils, multi-domain features are extracted, and physical health indicators and virtual health indicators are constructed. The physical health indicators include RMS, K, FE and three-layer wavelet packet node energy features (WPNE-01~WPNE-08), as shown in Figure 7. Virtual health indicators include principal component analysis (PCA-01~PCA-02) and MTS, as shown in Figure 8.

The above physical health indicators and virtual health indicators are used as preliminary health indicators. The monotonicity, trend and robustness of the above preliminary health indicators are evaluated, respectively. The results of each evaluation index of preliminary health indicators are shown in Table 2. Notably, the health indicators need to be normalized before the assessment. Equation (8) is as follows:(8)$$HI_{pre}^{test}(m)=(HI_{pre}(m)-HI_{pre}^{failure})/(HI_{pre}^{health}-HI_{pre}^{failure})$$
where $HI_{pre}^{health}$ is the preliminary health indicator of the electromagnetic coil health cycle; $HI_{pre}^{failure}$ is the preliminary health indicator of the previous cycle (failure cycle) of the fault cycle; $HI_{pre}(m)$ is the preliminary health indicator of the m-th cycle; and $HI_{pre}^{test}(m)$ is the preliminary health indicator for assessment after normalization.

In this section, an evaluation index based on a weighted linear combination of trend, monotonicity and robustness is constructed by (7). According to the importance of the three indexes in the coil insulation degradation process described in Section 2.3, the weights of the evaluation index J are set as $\omega_{1}=0.3,\omega_{2}=0.5,\omega_{3}=0.2$. As shown in (7), $\omega_{1}+\omega_{2}+\omega_{3}=1$. The value of the new evaluation index J is located at [0,1], and the closer to 1, the more sensitive the health indicator is. The results of the new evaluation index J for preliminary health indicators are shown in Figure 9. According to the selection criteria of the DSHI, the evaluation index value of HI14 is the closest to 1. Therefore, MTS is selected as a DSHI for the insulation degradation monitoring of electromagnetic coils based on the whole life cycle data in this section.

Normally, the MTS method requires the data distribution to follow a normal curve. If the data does not have a normal distribution, the calculation results of MTS may be biased. To solve the problem of data distribution, the Box-Cox power transformation [28,29] is employed in MTS to convert different data distributions into ones with similar characteristics. This ensures the same analytical methodology can be applied to various types of data, ultimately improving the accuracy of the analysis. The Box-Cox transformation can be used to transform variables with positive values that do not obey the normal distribution into variables that obey the normal distribution [30,31]. The assessment result of the transformed MTS is shown in Figure 10.

#### 3.1.3. Experiment Results Analysis of the Existing Methods

The resonant frequency-based method and Spearman correlation spectrum-based method are two existing classical impedance analysis techniques for insulation degradation monitoring. Therefore, coil impedance data obtained during the accelerating test is also analyzed by these two methods for comparison purpose. Wangkai et al. [32] found that the resonant frequency is influenced by the parasitic capacitance through computational analysis and can track the insulation degradation process of electromagnetic coils. In this paper, the resonant frequency of the whole life cycle is calculated by using reactance data. The result of coil insulation degradation monitoring based on resonant frequency is shown in Figure 11. The fluctuation of the resonant frequency from the healthy cycle to the five consecutive degradation cycles is minimal, and the amplitude change from the sixth degradation cycle to the final short-circuit fault cycle is significant.

Jameson N J et al. [19] used the Spearman correlation coefficient to find the frequency more relevant to the degradation process. The sensitive characteristics obtained by Spearman correlation coefficient are used to describe the degradation trend of the electromagnetic coil in this section. Figure 12 shows the reactance curve at the optimal frequency based on Spearman correlation coefficient. With the gradual aging of the coil, the reactance characteristics at the frequency closely associated with the aging cycle show a monotonous upward trend.

### 3.2. Twisted Pair Accelerated Test and Obtainment of Actual Turn Insulation Status

To find a benchmark that can represent the actual insulation health status of electromagnetic coils, a twisted pair accelerated fatigue test is designed to measure the breakdown voltage, which is a physical indicator that can truly reflect the degradation state of electromagnetic coils. To address the problem that it is difficult to measure the internal temperature of the transformer during the practical industrial applications, the relevant conditions of the accelerated aging test of the transformer are entered into the finite element simulation software for thermal simulation, and the temperature field distribution inside the transformer is obtained in this paper. The highest temperature inside the transformer is selected as the aging temperature of the twisted pair. The twisted pair samples are placed in a constant temperature aging chamber for accelerated fatigue testing, and then the breakdown voltage is measured.

The simulation process is as follows:Utilizing Catiav5R21, construct the geometric model of the transformer with consideration of its symmetry in the xyz direction. Only one-eighth of the transformer without symmetry is reserved for actual simulation to reduce unnecessary calculations. The geometric model should consist of the core, the core skeleton, the primary and secondary coils, as well as the insulation paper between them.Import the Catia 3D model into the ANSYS ICEM for meshing. Repair the geometric model first, block the components of the model, and then execute the corresponding mapping. Next, divide the model using a hexahedral mesh, and export the mesh file to prepare for the subsequent thermal simulation.Utilizing the Fluent module of Ansys, execute a thermal simulation on the model. Import the mesh file into the Fluent module of Ansys, and employ Fluent’s workflow to establish the physical properties of the material, including the thermal conductivity of each part. Among them, the coil and core are established to be anisotropic, the axis direction of the transformer coil skeleton is defined as Z direction, the thermal conductivity of this direction is 1.15 W/(m·K), the thermal conductivity of the coil in the X and Y directions is 230 W/(m·K). Additionally, set the boundary conditions, such as convection heat dissipation of the core and coil, as well as the ambient temperature, where the heat transfer coefficient is defined as 12 ${W/(m}^{2}\cdot K)$ and the ambient temperature is 80 °C. Finally, conduct the simulation under a 90 W overload condition.Analyze the results of the simulation. Based on the temperature distribution cloud of the model, identify the global maximum temperature and its location. The temperature field distribution is depicted in Figure 13, revealing that the highest temperature of 230 °C is located on the extension end of the transformer, where the intersection point of the primary and secondary coils can be found.

The twisted pair samples were subjected to thermal aging tests according to the IEC 60851-5 standard [15], as shown in Figure 14, with an aging temperature of 230 °C. After each aging cycle, 10 randomly selected specimens were used to measure the breakdown voltage at 50 Hz using the insulation withstand voltage tester HIOKI 3153, as shown in Figure 15. The mean value of the breakdown voltage for six thermal cycles (8 h each at 230 °C) is shown in Figure 16. During the gradual aging process of the twisted pair, the breakdown voltage gradually decreases.

### 3.3. Performance Analysis and Threshold Setting of the DSHI-Based Method

#### 3.3.1. Comparison Analysis of DSHI-Based Method with Existing Techniques

To compare the performance of DSHI-based precise insulation degradation monitoring method and the existing techniques, the breakdown voltage, the DSHI-based method (transformed MTS-based method), Spearman correlation coefficient-based method and resonance frequency-based method are normalized to obtain the degradation process of the whole life cycle of electromagnetic coils in the [0,1] interval, as shown in Figure 17. To quantify the degradation monitoring results, the indicator of degradation monitoring accuracy is defined, as shown in (9).
(9)$$\delta =1-\frac{1}{N}\sum_{i=1}^{N}\frac{\left|y_{i}-\bar{y_{i}}\right|}{y_{i}}\times 100\%$$
where δ represents the trend prediction accuracy of the degradation monitoring method, $y_{i}$ represents the value of the degradation monitoring method in the i-th cycle after normalization, $\bar{y_{i}}$ represents the value of the benchmark in the i-th cycle after normalization, and N represents the total number of cycles, i=1,2,…N.

Taking breakdown voltage as a benchmark, the degradation monitoring accuracy of the DSHI-based method, Spearman correlation coefficient-based method and resonant frequency-based method relative to the breakdown voltage is calculated. The calculation results show that the degradation monitoring accuracy of the DSHI-based method is 89.48%, which is 42.84% higher than that of resonant frequency-based method and 39.69% higher than that of Spearman correlation coefficient-based method. Therefore, the DSHI-based precise insulation degradation monitoring method has a higher accuracy than the existing mainstream methods.

#### 3.3.2. Threshold Setting of DSHI-Based Method

Setting the failure threshold properly for a health indicator is of great significance for degradation monitoring and predictive maintenance, because when the health indicator crosses its threshold, the coil should be replaced. Therefore, this section discusses the threshold setting for the DSHI-based precise insulation degradation monitoring method. Considering that MTS is determined as the DSHI in the experiment, the 3*σ* statistical approach is selected to set the threshold. 3σ is a commonly used method for anomaly detection. It considers the variability of the data and provides a more robust threshold for anomaly detection. According to the 3σ rule, the probability that the samples satisfying the normal distribution are distributed in [0.1] is 99.7%; then, the fault threshold is set as μ+3σ, where μ is the mean and σ is the standard deviation. The samples that exceed the fault threshold $threshold_{fault}$ are unhealthy samples.

Breakdown voltage measurement values during the accelerated test is used to verify the effectiveness of the DSHI threshold setting. Considering that as the breakdown voltage measured gradually approaches the rated voltage, the insulation failure probability increases accordingly, insulation failure probability model is defined as (10).
(10)$$p(t)=1-\exp{\left(-\frac{BV_{measure}(t)-BV_{health}}{V_{rated}-BV_{health}}\right)}^{3}$$
where p(t) is the insulation failure probability, $BV_{measure}(t)$ denotes the breakdown voltage measured at the t-th aging cycle, and $V_{rated}$ denotes the rated voltage. $BV_{health}$ denotes the breakdown voltage of healthy coils. In detail, the measured breakdown voltage values are shown in Figure 16, where the breakdown voltage value $BV_{health}$ in the healthy state is 620 V, and the $V_{rated}$ in this test is 220 V.

The insulation probability calculation results during the accelerated test are shown in Table 3. According to the experiment results mentioned in Figure 16, the coil failed in the seventh cycle, while the probability by Equation (10) is 89.46%, which verified the proposed insulation failure probability model based on breakdown voltage. Further, the threshold of insulation failure probability model based on breakdown voltage is defined as 85% in this paper.

Figure 18 shows the evolution trend comparison for DSHI, and the insulation failure probability calculated by (10), where the red solid line indicates the DSHI, and the black solid line indicates insulation failure probability model based on breakdown voltage. The red dashed line is the threshold for DSHI, and the black dashed line indicates the threshold for insulation failure probability model based on breakdown voltage. Both agree that the coil under test should be replaced after the sixth aging cycle, which proves that the threshold setting method of the DSHI is effective.

## 4. Conclusions

Insulation degradation monitoring plays an important role in avoiding the unexpected shutdown of electrical equipment, especially in safety-critical domains. This paper proposed an insulation degradation monitoring method by exploring the macroscopic and microscopic effects of coil thermal aging. At the macro level, an evaluation index based on a weighted linear combination of trend, monotonicity and robustness is proposed to construct DSHI based on the coil high-frequency response parameters during the whole life cycle. The simultaneous development of coil FEA and twisted pair accelerated tests are performed at the micro level. Comparison analysis of macro and micro results shows that the degradation monitoring accuracy of the DSHI-based method is 89.48%, which is higher than mainstream insulation degradation methods, like the resonant frequency-based method and Spearman correlation coefficient-based method. Further, a breakdown voltage-based insulation failure probability model is defined to verify the threshold setting of the DSHI-based method. The proposed method supports the development of a coil insulation health assessment model that enables electrical equipment condition-based maintenance. Considering that the effect of temperature on high-frequency response parameters measurement is not considered in this paper, in the future research, a high-frequency response parameter mapping model at different measurement temperatures will be developed to improve the robustness of the proposed method.

## Figures and Tables

**Figure 1 entropy-25-01354-f001:**
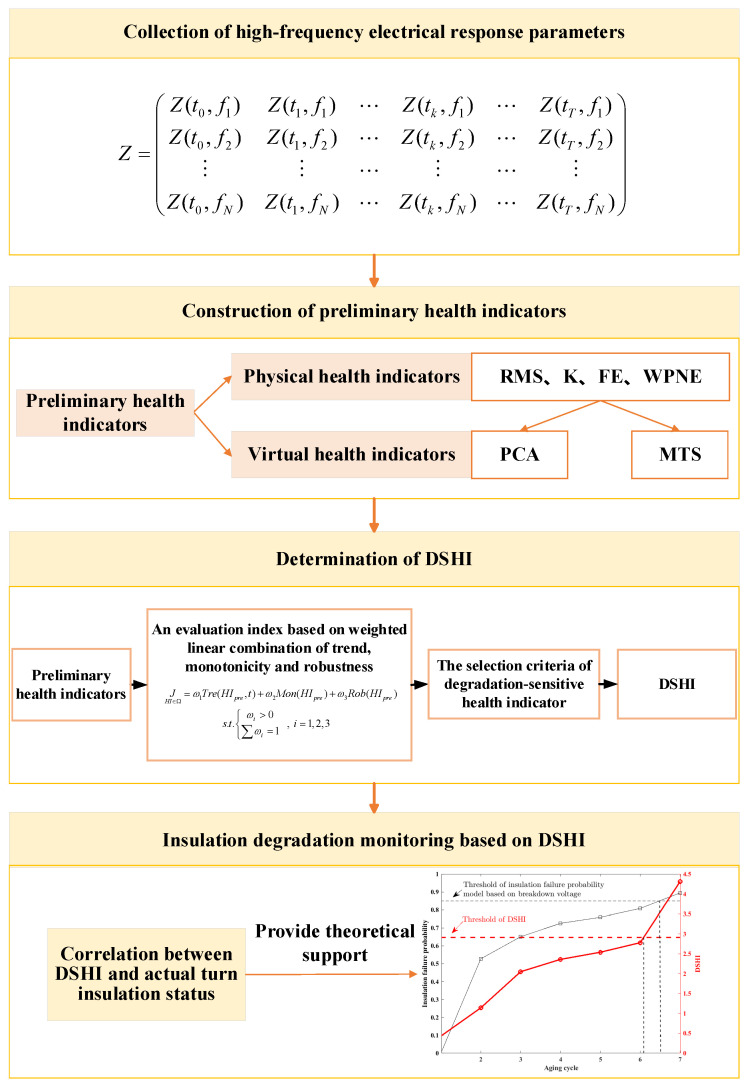
The methodology framework of coil precise insulation degradation monitoring.

**Figure 2 entropy-25-01354-f002:**
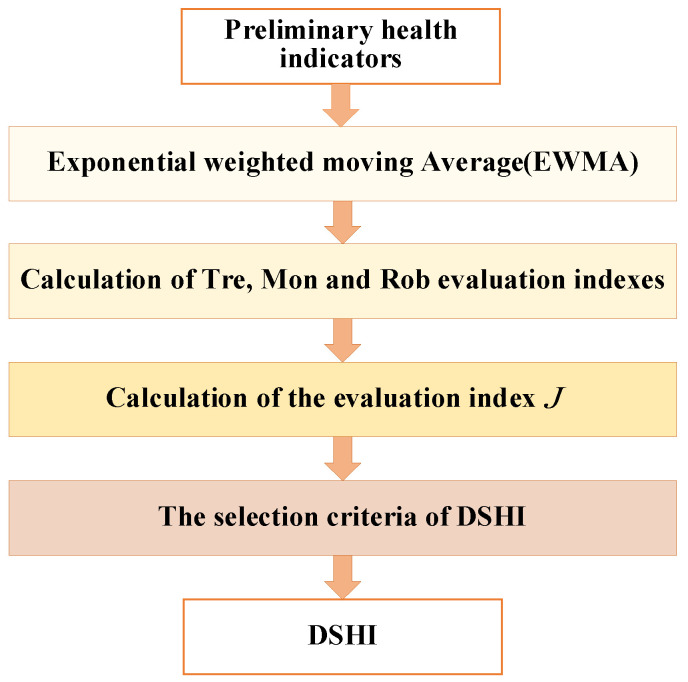
The DSHI determination process.

**Figure 3 entropy-25-01354-f003:**
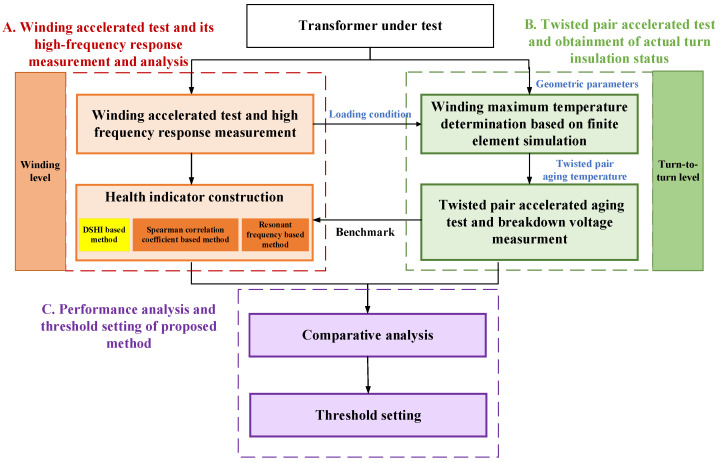
Experimental scheme.

**Figure 4 entropy-25-01354-f004:**
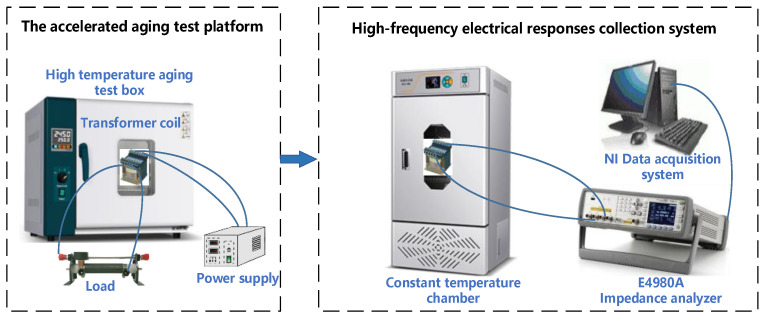
The accelerated aging test platform and high-frequency electrical responses collection system.

**Figure 5 entropy-25-01354-f005:**
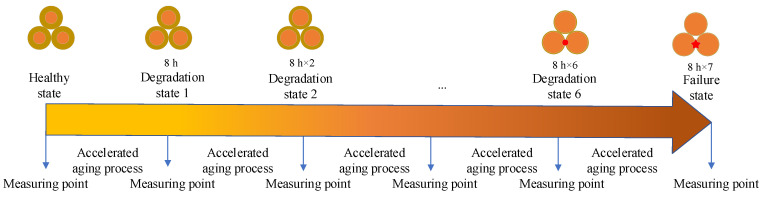
The accelerated fatigue test procedure.

**Figure 6 entropy-25-01354-f006:**
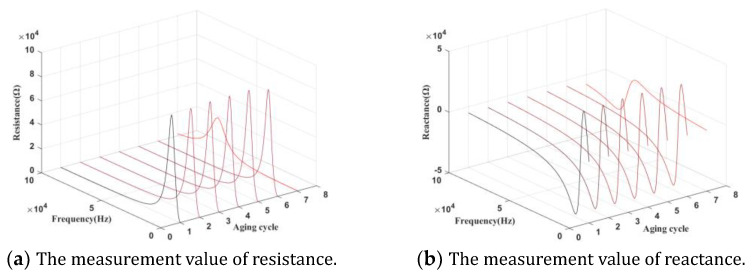
Resistance and reactance for the whole life cycle of electromagnetic coils.

**Figure 7 entropy-25-01354-f007:**
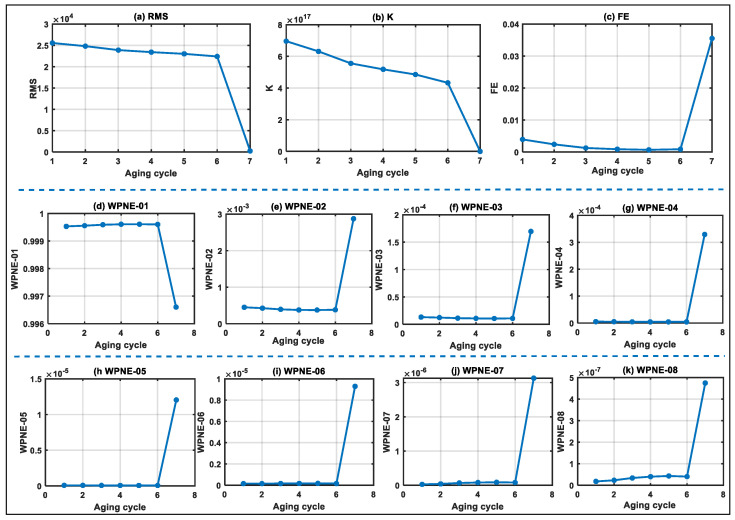
Physical health indicators of electromagnetic coils in the whole life cycle.

**Figure 8 entropy-25-01354-f008:**
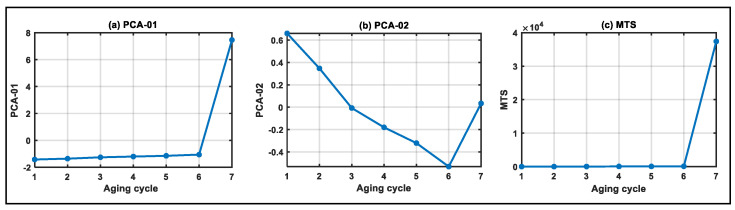
Virtual health indicators of electromagnetic coils in the whole life cycle.

**Figure 9 entropy-25-01354-f009:**
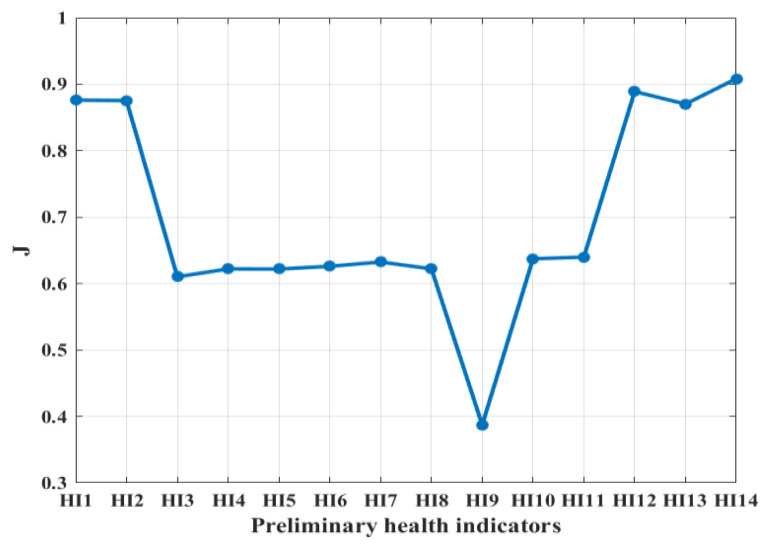
The results of the evaluation index J for preliminary health indicators.

**Figure 10 entropy-25-01354-f010:**
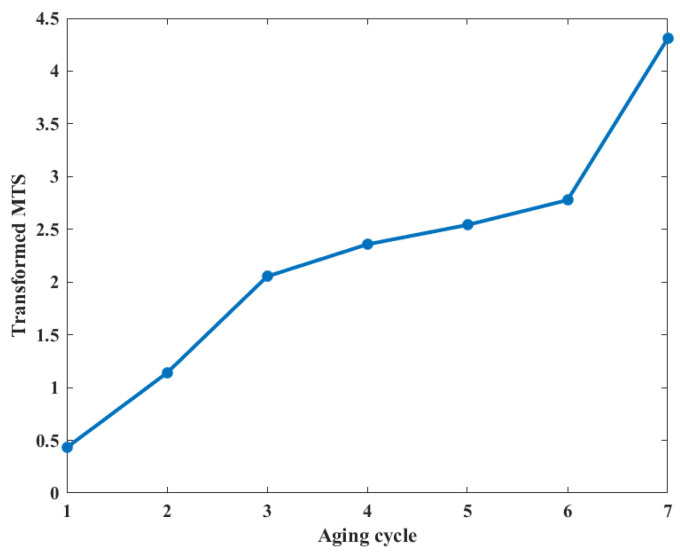
The result of coil insulation degradation monitoring based on transformed MTS.

**Figure 11 entropy-25-01354-f011:**
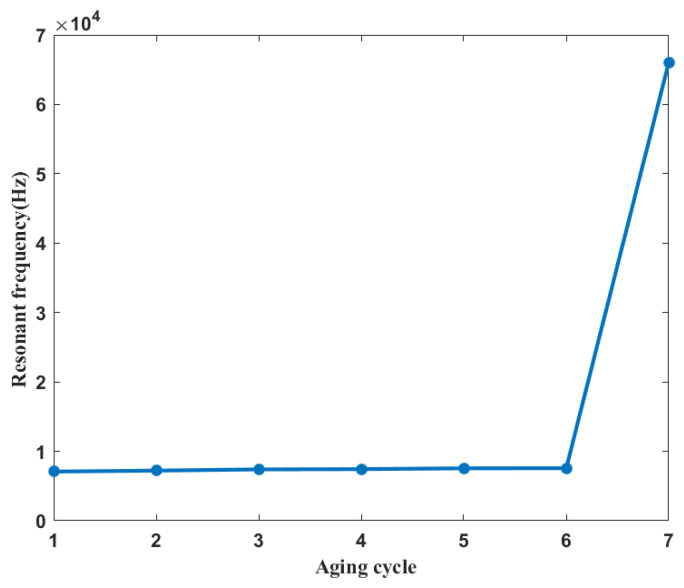
The result of coil insulation degradation monitoring based on resonant frequency.

**Figure 12 entropy-25-01354-f012:**
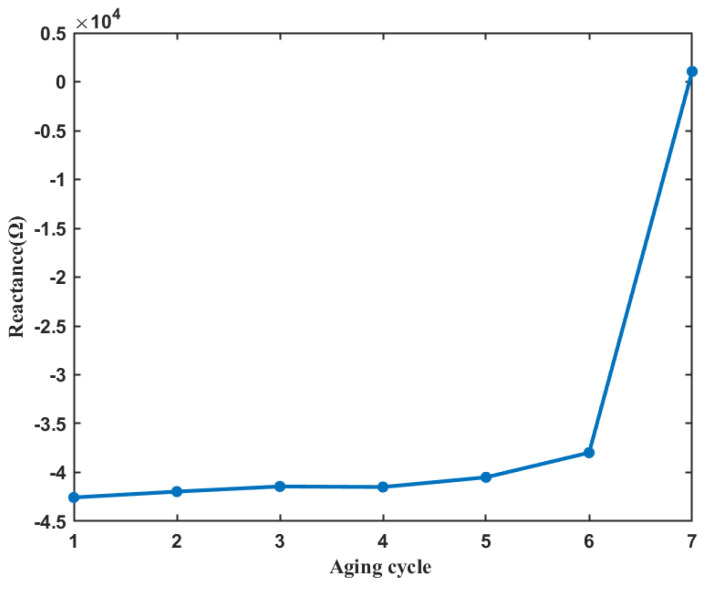
The result of coil insulation degradation monitoring based on Spearman correlation coefficient.

**Figure 13 entropy-25-01354-f013:**
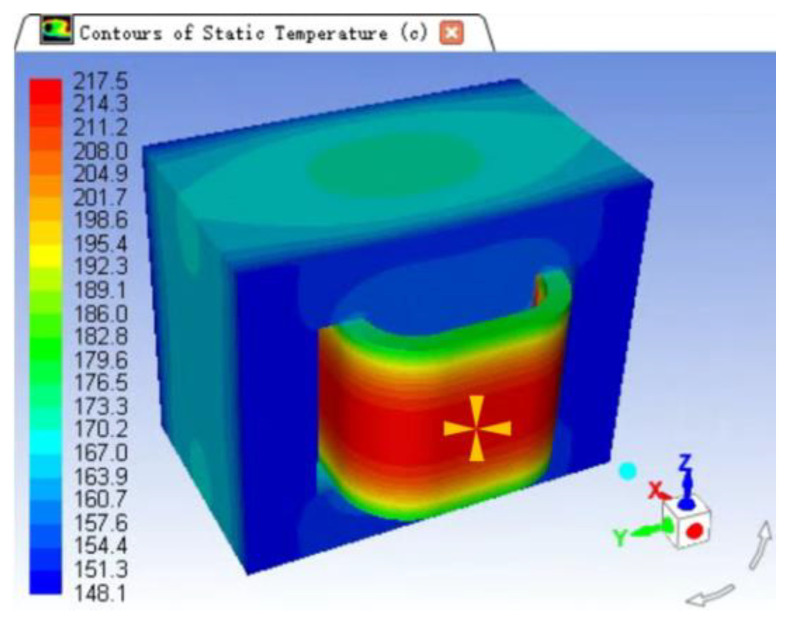
Transformer thermal simulation results.

**Figure 14 entropy-25-01354-f014:**
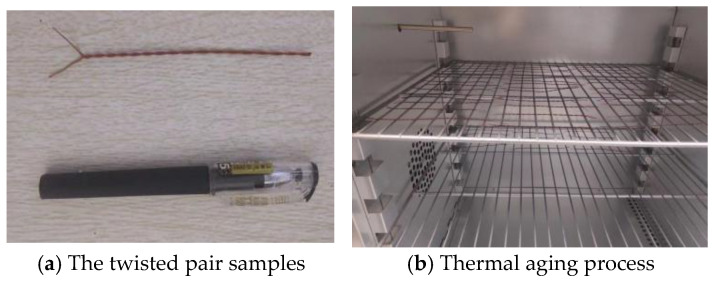
Thermal aging test of the twisted pair.

**Figure 15 entropy-25-01354-f015:**
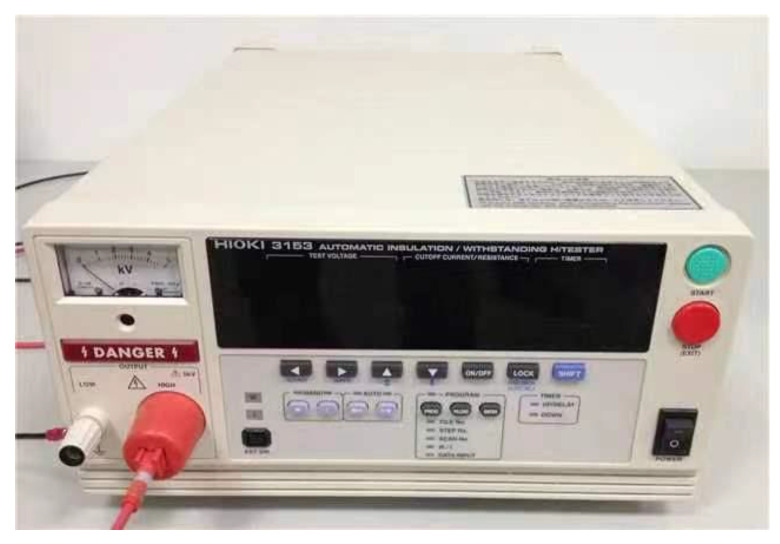
Insulation withstand voltage tester.

**Figure 16 entropy-25-01354-f016:**
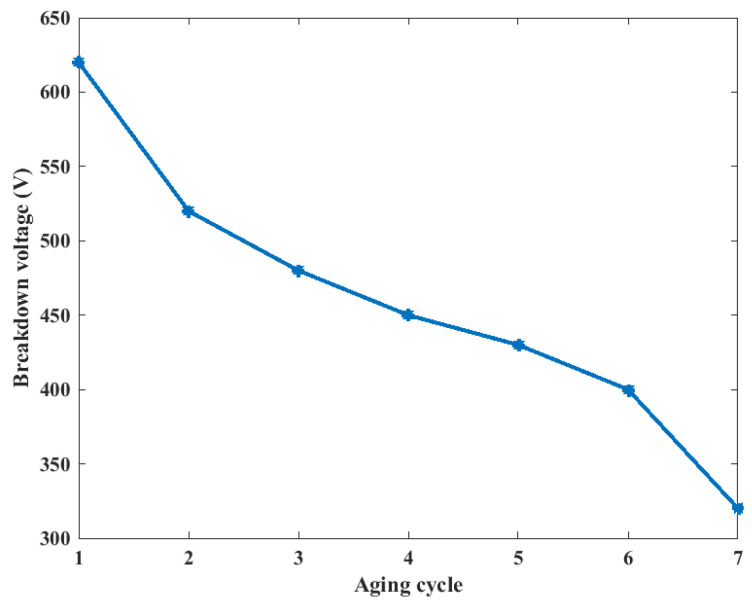
The variation of the mean value of the breakdown voltage during the aging process.

**Figure 17 entropy-25-01354-f017:**
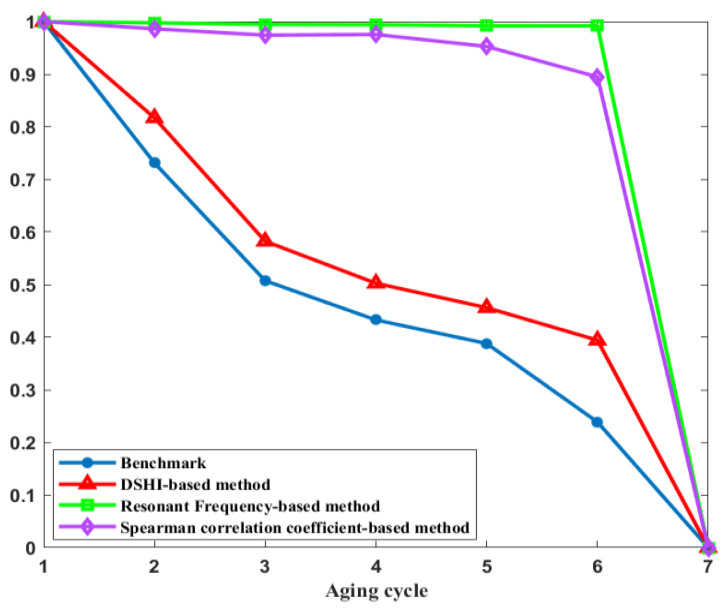
Comparison of different degradation monitoring methods.

**Figure 18 entropy-25-01354-f018:**
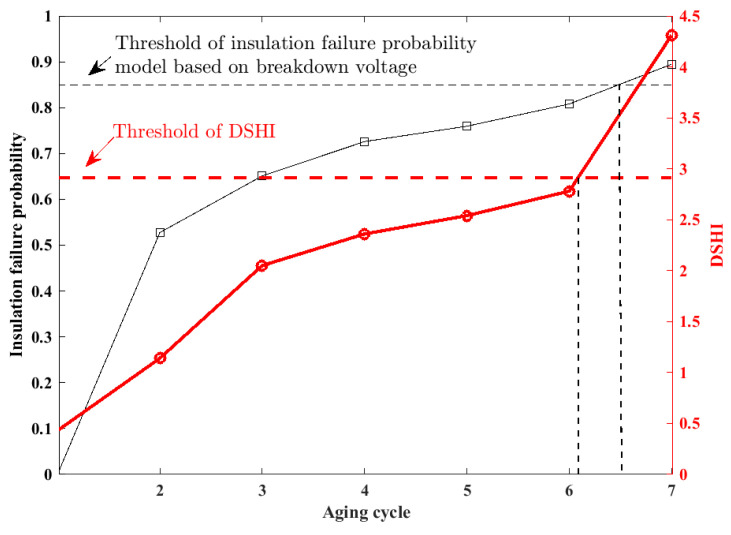
Evolution trend Comparison for DSHI and insulation failure probability based on Breakdown voltage.

**Table 1 entropy-25-01354-t001:** The calculation equations of preliminary health indicators.

Preliminary Health Indicators		Equation
Physical health indicators	Root mean square (RMS)	$HI_{rms}=\sqrt{\frac{1}{N}\sum Z{\left(t_{k}\right)}^{2}}$
	Kurtosis (K)	$HI_{k}=\frac{1}{HI_{rms}{}^{4}}\sum_{}(Z(t_{k})-\bar{Z(t_{k})}{)}^{4}$
	Fuzzy entropy (FE)	$HI_{fe}=\ln\phi^{m}(n,r)-\ln\phi^{m+1}(n,r)$
	Wavelet packet node energy (WPNE)	$HI_{wpne_{i}}=\frac{1}{\sum_{i}z_{i}{}^{2}}z_{i}{}^{2}$
Virtual health indicators	Principal component analysis (PCA)	$HI_{pca}=PZ(t_{k})$
	Mahalanobis-Taguchi system (MTS)	$HI_{MD_{j}}=\frac{1}{p}S_{j}\cdot corr^{-1}\cdot S_{j}{}^{T}$

**Table 2 entropy-25-01354-t002:** Preliminary health indicator evaluation indexes.

Preliminary Health Indicators		Evaluation Indexes			Symbols
		Trend	Monotonicity	Robustness	
Physical health indicators	RMS	0.9904	1	0.3957	HI1
	K	0.9894	1	0.3930	HI2
	FE	0.8776	0.6	0.2346	HI3
	WPNE-01	0.8925	0.6	0.2717	HI4
	WPNE-02	0.8917	0.6	0.2717	HI5
	WPNE-03	0.9034	0.6	0.2747	HI6
	WPNE-04	0.9247	0.6	0.2755	HI7
	WPNE-05	0.8607	0.6	0.3199	HI8
	WPNE-06	0.7271	0.2	0.3459	HI9
	WPNE-07	0.9212	0.6	0.3054	HI10
	WPNE-08	0.9246	0.6	0.3119	HI11
Virtual health indicators	PCA-01	0.9965	1	0.4505	HI12
	PCA-02	0.9851	1	0.3731	HI13
	MTS	0.9775	1	0.5759	HI14

**Table 3 entropy-25-01354-t003:** Coil insulation failure probability for the whole life cycle.

Cycle	1	2	3	4	5	6	7
p(t)	0	52.76%	65.01%	72.06%	75.95%	80.8%	89.46%

## Data Availability

Not applicable.

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
