# Peer review of "Degradation-Sensitive Health Indicator Construction for Precise Insulation Degradation Monitoring of Electromagnetic Coils"

_entropy, 2023, doi:10.3390/e25091354_

Round 1
Reviewer 1 Report
In this paper, an evaluation index based on weighted linear combination of trend monotonicity and robustness is proposed to construct degradation-sensitive health indicator based on high-frequency electrical response parameters for precise insulation degradation monitoring. It is of interest. However, the authors should address the following points to further improve the quality.
1. The contributions of this paper are not clear. They should be highlighted at the end of the introduction. In addition, more information on the background and problem statement should be added.
2. The quality of figures must be improved.
3. How to determine the values of the three weights in Eq. (7).
4. It is better to give a detailed procedure to describe the whole framework of the proposed approach.
5. Literature review on the deep learning-based methods of health monitoring is limited. The authors may be benefited by reviewing more papers such as 10.1016/j.engappai.2023.106927 and 10.1109/TMECH.2023.3300359.
6. The validation is a big concern in this work. How authors can avoid bias validation? Authors can perform a significant test to show the efficiency of the method.
7. The linguistic quality needs improvement. I can find some typos.
The linguistic quality needs improvement. I can find some typos.
Reviewer 2 Report
A very well-structured manuscript focused on insulation degradation monitoring of electromagnetic coils. Congrats to the authors.
State-of-the art is relates the nowadays solution and the research interest area, also the references are the majority of the past 10 years which reveals the up-to-date research. Anyway, can be improved with some other research focused on windings degradation.
Good simulation support for the transformer model proposed, authors are using up-to-date simulation software.
The work is based on a very well-done expert interpretation of the resulting data.
The model proposed can be useful for future researchers in order to anticipate insulation degradation of electrical machines due to electric-thermal wearing.
Good mathematical support for the simulation proposed. Anyway, can be improved with some deep analysis but is enough for the scope proposed. In the future research can improve the analytic cases.
The paper is interesting and can be published in the journal with minor changes, regarding article structure form according to the journal template. For example, figures 7 and 8 are not easy to read…
Regards to the authors,
Reviewer
Round 2
Reviewer 1 Report
The revision has addressed all my issues. The linguistic quality needs improvement.
The linguistic quality needs improvement.
Reviewer 2 Report
The authors have replied to all the queries and concerns with proper references. However, to improve the connectivity for readers, authors can develop their research in future articles by using of better approach regarding experiments.
Good luck with future research!